

# A Time-Dependent Three-Dimensional Magnetopause Model Based on Quasi-elastodynamic Theory

Yaxin Gu[1,2], Yi Wang[1*], Fengsi Wei[1], Xueshang Feng[1], Andrey Samsonov[2], Xiaojian Song[3], Boyi Wang[1], Pingbing Zuo[1], Chaowei Jiang[1], Yalan Chen[1], Xiaojun Xu[4], Zilu Zhou[4]

Shenzhen Key Laboratory of Numerical Prediction for Space Storm, College of Aerospace Science and Technology, Harbin Institute of Technology, Shenzhen, 518055, China
Mullard Space Science Laboratory, University College London, Dorking, RH56NT, United Kingdom
Shandong High Technology Research, Shandong, 250100, China
State Key Laboratory of Lunar and Planetary Sciences, Macau University of Science and Technology, Macau, 999078, China

*Correspondence to*: Yi Wang (wingwy@mail.ustc.edu.cn)

**Abstract.** The interaction between the solar wind and Earth's magnetosphere is a critical area of research in space weather and space physics. Accurate determination of the magnetopause position is essential for understanding magnetospheric dynamics. While numerous magnetopause models have been developed over past decades, most are time-independent, limiting their ability to elucidate the dynamic movement of the magnetopause under varying solar wind conditions. This study introduces the first time-dependent three-dimensional magnetopause model based on quasi-elastodynamic theory, named the POS (Position-Oscillation-Surface wave) model. Unlike existing time-independent models, the POS model physically reflects the dynamic responses of magnetopause position and shape to time-varying solar wind conditions. The predictive accuracy of the POS model was evaluated by using 38,887 observed magnetopause crossing events. The model achieved a root-mean-square error of 0.768 Earth radii ($R_E$), representing a 18.7% improvement over five widely used magnetopause models. Notably, the POS model demonstrated superior accuracy under highly disturbed solar wind conditions (24.9% better) and in higher latitude regions (28.7% better) and flank regions (35.2% better) of the magnetopause. The POS model's remarkable accuracy, concise formulation, and fast computational speed enhance our ability to predict magnetopause position and shape in real-time. This advancement is significant for understanding the physical mechanisms of space weather phenomena and improving





the accuracy of space weather forecasts. Furthermore, this model may provide new insights and
methodologies for constructing magnetopause models for other planets.

## 1 Introduction

The magnetopause, the boundary between the interplanetary magnetic field (IMF) and Earth's
magnetic field, plays a crucial role in space weather forecasting and understanding solar wind-
magnetosphere coupling mechanisms (Willis, 1971; P. Song, 1996; Russell, 2003). It acts as a
protective shield against hazardous energetic particles while simultaneously serving as the primary
interaction region for solar wind-magnetosphere coupling. The magnetopause exhibits considerable
dynamic behaviour due to continuous solar wind variations and various instabilities, even under steady
solar wind conditions (Anderson et al., 1968; Song et al., 1988; Eastwood et al., 2015). These dynamics
can lead to radiation belt particle loss, field-aligned current intensification, ultra-low frequency wave
generation, and solar wind energy conversion into the radiation belts, polar regions, and ionosphere
(Haerendel, 1990; Mann et al., 2012; Plaschke, 2016; Mottez, 2016; Archer et al., 2019). Consequently,
comprehending the interactions between solar wind and magnetopause is vital for advancing
magnetosphere dynamics and improving space weather prediction capabilities (Feng, 2020; Zong et
al., 2020).
Numerous magnetopause models have been established over the past few decades, generally
categorized as physical (or principal) models (Ferraro, 1952; Beard, 1960; Spreiter et al., 1966) and
empirical models (Fairfield, 1971; Tsyganenko, 1989; Shue et al., 1998; Lin et al., 2010). Physical
models are primarily based on the classic Chapman-Ferraro theory proposed in the 1930s (Chapman
and Ferraro, 1930), which states that the magnetopause's equilibrium position is determined by the
pressure balance between solar wind dynamic pressure ($P_{dyn}$) and magnetospheric magnetic pressure
($P_b$). Since the 1960s, the launch of numerous satellites has provided us with a large number of samples
of magnetopause crossing events (MCEs), thereby creating the possibility for the establishment of
empirical models (Fairfield, 1971; Sibeck, 1991; Petrinec and Russell, 1996; Shue et al., 1998; Lin et



al., 2010). Many empirical models rely on two key parameters, $P_{dyn}$ and IMF $B_z$, and some of them
include the Earth's dipole tilt angle (Φ) to calibrate the higher latitude zone. Besides, some empirical
models, proposed from the 1980s, combine physical processes of solar wind-magnetosphere
interactions with satellite observation fitting and involved the impact of magnetospheric currents
system (Tsyganenko, 1989, 1996). Regardless of the assumptions on which these models are based,
all these models have contributed to our understanding of magnetopause movement and its response
to solar wind conditions, in particular, many of them have been widely used in the prediction of the
magnetopause due to their simple form and high prediction accuracy.
However, it should be aware that these models primarily describe the average steady-state
characteristics of the magnetosphere. To accurately describe the dynamic coupling process of solar
wind-magnetosphere interaction, it is essential to incorporate time partial derivatives into the dynamic
equations (Smit, 1968; Petrinec, 2001; Borovsky and Alejandro Valdivia, 2018). This approach,
however, complicates the solution of model equations, often necessitating numerical simulations such
as magnetohydrodynamics (MHD) (Powell et al., 1999; Raeder et al., 2001; Lyon et al., 2004; Tóth et
al., 2005; Merkin and Lyon, 2010), particle-in-cell (PIC) (Moritaka et al., 2012; Ashida et al., 2014;
Walker et al., 2019), and hybrid simulations (Gargaté et al., 2008; Omelchenko et al., 2021; Ala-Lahti
et al., 2022). Numerical simulations are widely used in exploring solar wind-magnetosphere coupling
and can accurately reveal the position of the magnetopause changing with the time-varying solar wind.
Their prediction accuracy is generally much better than the magnetopause model mentioned above.
However, the introduction of time partial derivatives makes equations very difficult to solve. In
addition, many prominent numerical simulation models may not include properly all magnetospheric
current systems (e.g. the ring current or the magnetospheric-ionospheric currents), therefore this may
result in systematic errors of the magnetopause prediction (Samsonov et al., 2016). Moreover,
numerical models are solved on supercomputers, consuming a significant amount of computing
resources and time, rendering them impractical for real-time space weather forecasting (Raeder et al.,
2001; Lyon et al., 2004; Tóth et al., 2005; Feng, 2020). This limitation highlights the need for more



efficient, yet accurate, magnetopause models that can capture the dynamic nature of the magnetopause
while remaining computationally feasible for real-time applications. Such models would significantly
enhance our ability to predict and understand space weather phenomena, bridging the gap between
theoretical understanding and practical forecasting capabilities.
Apart from numerical simulations, very few time-dependent magnetopause models have been
historically developed (Smit, 1968; Freeman et al., 1995; Børve et al., 2011). Figure 1 illustrates the
fundamental difference between time-independent and time-dependent models. In time-independent
models, the magnetopause position is directly correlated with instantaneous solar wind conditions. For
example, a step-like increase in solar wind dynamic pressure (such as a shock) corresponds to an
immediate step-like compression of the magnetopause (Figure 1a). However, this simplification fails
to capture the real dynamics of the magnetopause. In reality, the magnetopause undergoes a more
complex process of compression and recovery, exhibiting oscillatory characteristics in response to
abrupt changes in solar wind conditions, as shown in Figure 1b (Freeman and Farrugia, 1998; Hu et
al., 2005; Desai et al., 2021). Time-dependent models aim to capture these dynamic processes,
providing a more accurate representation of magnetopause behaviour. To describe these dynamic
responses, it is necessary to incorporate time partial derivatives into the governing equations. However,
this inclusion significantly complicates the solution process. Consequently, existing time-dependent
models are predominantly one-dimensional and remain in a preliminary stage of development.

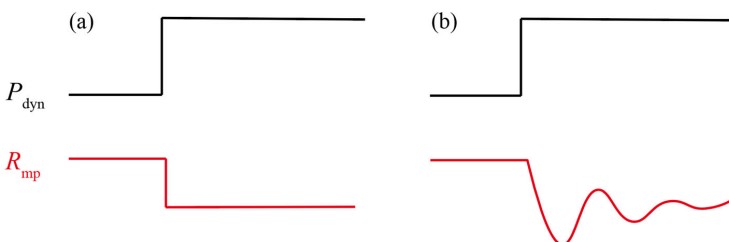


**Figure 1 The schematic diagram of time-independent (a) and time-dependent (b) magnetopause models.**



Smit (1968) conceptualized the magnetopause as a rigid surface and attempted to explain its
motion from the perspective of periodic vibration; Freeman et al. (1995) investigated the influence of
inertial and damping effect on the magnetosphere, employing magnetohydrodynamics to analyse the
magnetopause motion; Børve et al. (2011) set up a non-adjustable model to analyse the oscillation
period of the magnetopause. By investigating the movement of the subsolar point in response to time-
varying solar wind, these models are primarily constructed to elucidate specific physical phenomena
linked to solar wind-magnetosphere interaction, yet they lack the capability to provide a real-time
depiction of the three-dimensional magnetopause position and shape.
Hence, the challenge of constructing time-dependent models lies in balancing the need for
accurate dynamic representation with computational feasibility. Although time-dependent models
offer a more realistic depiction of magnetopause behaviour, their complexity has limited their
development and application, particularly in three-dimensional space. This highlights the necessity for
new strategies that can capture time-dependent dynamics while ensuring practical utility for
computation, especially for real-time space weather forecasting and related magnetospheric researches.
Previously,  our work revealed the quasi-elastodynamic processes involved in the interaction between
solar wind and magnetosphere (Gu et al., 2023). It suggests that the dynamic behaviour of each point
on the magnetopause can be viewed as an equilibrium position (P), radial global oscillations around
equilibrium position (O), and surface wave-like structure around the flank regions (S). This work offers
a practical framework for developing a time-dependent three-dimensional magnetopause model.
However, our previous work primarily focused on elucidating the quasi-elastic process, with less
emphasis on the outcomes of model predictions (Gu et al., 2023). Key factors influencing
magnetopause dynamics, such as the IMF $\boldsymbol{B}_z$ and Earth's dipole tilt angle ($\Phi$), were not incorporated.
Additionally, the adjustable parameters in the equations were simply chosen and lack of thorough
calibrations. Moreover, both our prior work and most published magnetopause models (Petrinec and
Russell, 1996; Shue et al., 1998; Gu et al., 2023) relied on a relatively limited dataset of low-latitude
satellite observations, leading to constraints in accurately representing the higher latitude and flank



regions of the magnetopause. To address these limitations and overcome the inherent shortcomings of
time-independent models, particularly their inability to reflect the dynamic responses of the
magnetopause position and shape to time-varying solar wind conditions, we propose a time-dependent
three-dimensional magnetopause model. This model, which has been tested with the largest dataset of
MCEs to date (38,887 events), demonstrates remarkable prediction accuracy compared to five widely
used magnetopause models. Besides, it offers unparalleled real-time computation speed and a concise
form relative to numerical simulations. We have named this model the Position-Oscillation-Surface
wave (POS) model.
**2 Dataset and other magnetopause models for comparison**
The THEMIS (Time History of Events and Macroscale Interactions during Substorms) mission
(Angelopoulos, 2008), which consists of five spacecrafts launched into similar elliptical, near-
equatorial orbits in 2007, has significantly enhanced our ability to observe the magnetosphere. The
mission provides high-resolution (~3 s) magnetic field measurements through the THEMIS/Flux Gate
Magnetometer (FGM) (Auster et al., 2008) and plasma data from the THEMIS/electrostatic analyser
(ESA) (Mcfadden et al., 2008). The Cluster II mission (Escoubet et al., 2001), involving four identical
spacecraft launched in 2000, also offers high-resolution (~4 s) magnetic field measurements using the
CLUSTER/Flux Gate Magnetometer (FGM) (Balogh et al., 1997) and particle data and moments from
the Cluster Ion Spectrometry Hot Ion Analyser (CIS-HIA) (RÈme et al., 1997).
The WIND spacecraft, launched into orbit around Earth in 1994 and relocated to Lagrange L1
point after 2004, provides reliable, high-quality in situ measurements of the solar wind. This study
utilizes high-resolution (~3 s) plasma data from the WIND/3D Plasma Analyzer (3DP) (Lin et al.,
1995) and magnetic field data from the Magnetic Field Investigation (MFI) (Lepping et al., 1995) for
upstream solar wind observations. For this study, we have compiled a dataset consisting of 51,590
THEMIS MCEs and 38,321 Cluster MCEs. After excluding redundant, invalid data and nightside
MCEs ($X_{GSM} < 0$ $R_E$), a total of 38,018 THEMIS MCEs and 869 CLUSTER MCEs (see Figure 2) are



matched with upstream solar wind observations. The time shift ($\Delta t$) from WIND to each MCE was
determined by comparing the time of each magnetopause crossing ($t_1$) with the estimated arrival time
of the corresponding solar wind observation from WIND ($t_0 + \Delta t$). This condition was satisfied when
($t_0 + \Delta t$) – $t_1$ < 300 s, where $\Delta t$ was calculated using the formula $\Delta t = (L1 - r) / <v_x>$. Here, $<v_x>$
represents the 1-hour sliding average of the solar wind velocity's x-component as observed by WIND
at L1 (L1 = 235 $R_E$), and r is the radial position of the magnetopause derived from each MCE. A
summary of these events is provided in Table 1. The distribution of matched solar wind conditions for
MCEs is shown in Figure 3. All the data is available in the CDAWeb database
(http://cdaweb.gsfc.nasa.gov/), and the time resolution of the magnetic field and plasma data used in
the study is interpolated into 3 seconds, set in GSM coordinates.

**Table 1 Summary of collected 89,911 satellite MCEs and dataset used in this paper**

| Dataset | Satellite | Time interval | Number of datasets |
|---|---|---|---|
| Song et al. (2021) | THEMIS | 2007-2022 | 17,647 |
| Staples et al. (2020a) | THEMIS | 2007-2016 | 33,943 |
| Grimmich (2024) | CLUSTER | 2001-2020 | 38,321 |
| In this paper | THEMIS/CLUSTER | 2004-2022 | 38,887 |

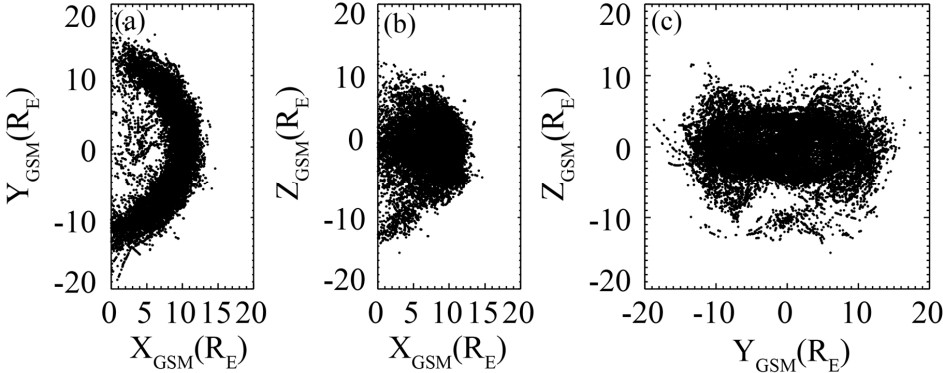


**Figure 2 Projections of 38,887 MCEs in GSM coordinate: (a) X-Y plane, (b) X-Z plane, (c) Y-Z plane.**



In previous research on time-independent magnetopause models, the physical models (Ferraro,
1952; Beard, 1960; Spreiter et al., 1966), although theoretically grounded, usually oversimplify
intricate solar wind-magnetosphere interactions to facilitate calculations, usually without
demonstrating apparent higher prediction accuracy compared to widely-used empirical models
(Petrinec and Russell, 1996; Shue et al., 1997; Shue et al., 1998; Chao et al., 2002; Lin et al., 2010)..
Hence, this article concentrates on comparing several notable time-independent empirical models
renowned for their superior prediction accuracy (Petrinec and Russell, 1996; Shue et al., 1997; Shue
et al., 1998; Chao et al., 2002; Lin et al., 2010).

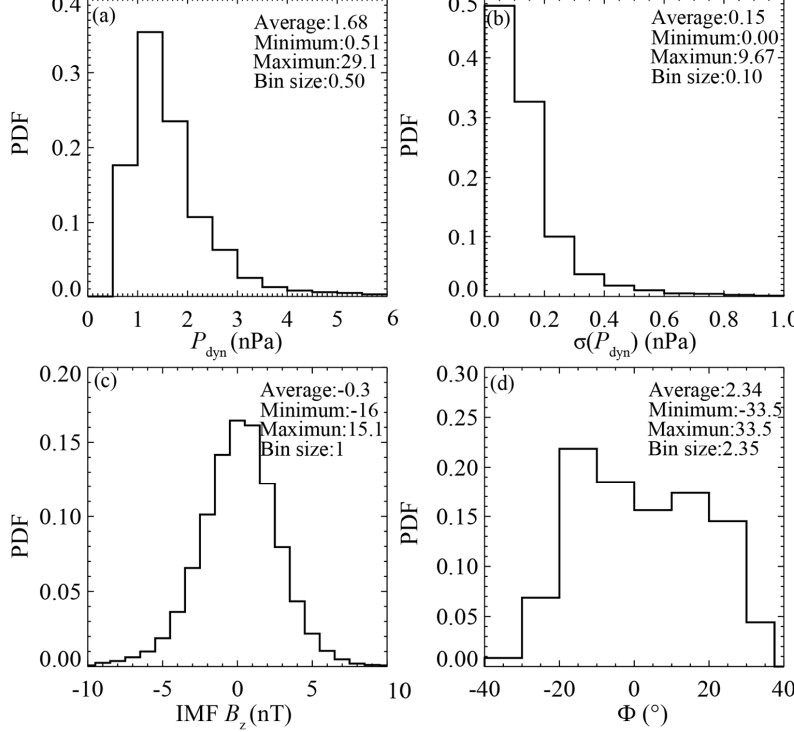


**Figure 3  Probability density function of the upstream solar wind observation in: (a) solar wind dynamic**
**pressure $P_{dyn}$ ,(b) standard deviation of dynamic pressure ($P_{dyn)}$, (c) interplanetary magnetic field $B_z$**
**component (IMF $B_z$) and (d) dipole tilt angle (Φ).**




Empirical models are typically constructed using satellite observations of MCEs. While these
models vary in their use of satellite datasets, parameters considered, coordinate systems employed,
and functions applied, most are parameterized using the dynamic pressure ($P_{dyn}$) and the interplanetary
magnetic field $B_z$ component (IMF $B_z$). For example, Petrinec and Russell (1996) (hereafter PR96)
employed an ellipsoidal function to construct a magnetopause model, while Shue et al. (1997)
(hereafter S97) developed a flexible function incorporating two variables: the subsolar magnetopause
position ($R_0$) and the tail flaring angle ($\alpha$). This function has gained widespread use as a foundational
approach to describing magnetopause shape. For instance, Shue et al. (1998) (hereafter S98) accounted
for the saturation effect of IMF $B_z$ on $R_0$, and Chao et al. (2002) (hereafter C02) extended their model
for application under normal and extreme solar wind conditions.
Nevertheless, these models primarily rely on low-latitude satellite observations and may not
adequately capture the distinctive characteristics of the magnetopause in the higher latitude region.
Besides, they are constructed with $P_{dyn}$ and IMF $B_z$ , while it is found that the dipole tilt angle $\Phi$ is of
great significance in modelling magnetopause, especially in the higher latitude region. Formisano et
al. (1979) constructed an average magnetopause size and shape for two dipole tilt angle values ($\Phi$).
Boardsen et al. (2000) developed a higher latitude magnetopause model parameterized by not only
$P_{dyn}$ and IMF $B_z$ but also the dipole tilt angle ($\Phi$), recognizing its significant influence on the shape of
the higher latitude magnetopause. While this model is specifically designed for higher latitude regions,
it is not as effective in accurately calculating the magnetopause at low latitudes compared to other
models due to inherent limitations.
The above models are generally developed under the assumption of axial symmetry, while the
actual magnetopause shape is asymmetric in both the Y and Z directions, so they are essentially 2D or
2.5D models. To describe the 3D structure of the magnetopause, Lin et al. (2010) (hereafter L10)
developed a three-dimensional magnetopause model parameterized by $P_{dyn}$, thermal pressure ($P_t$), IMF
$B_z$, and $\Phi$. The coordinate systems employed in these empirical models are typically in aberrated
coordinates which accounts for Earth's orbital motion (Petrinec and Russell, 1996; Shue et al., 1997;



Shue et al., 1998; Chao et al., 2002), or the corrected coordinates which compensates for both Earth's
orbital motion and deviations in solar wind velocity from the Sun-Earth line (Boardsen et al., 2000;
Lin et al., 2010). A summary of the five widely used magnetopause models is presented in Table 2.
**Table 2 Summary of five widely used magnetopause models and POS model**

| Model Name | Number of (higher latitude) MCEs used | Time range of MCEs used | Dimensions |
|---|---|---|---|
| PR96 | 1,147 | 1979-1980 | 2D/2.5D |
| S97 | 553 | 1978-1986 | 2D/2.5D |
| S98 | 553 | 1978-1986 | 2D/2.5D |
| C02 | 552 | 1978-1986 | 2D/2.5D |
| L10 | 1,226 (1,482) | 1994-2008 | 3D |
| POS | 31,562 (7,325) | 2004-2022 | 3D |

**3 The POS Model**
In our previous work (Gu et al., 2023), we modelled the compression-recovery process of the
magnetopause as a quasi-elastodynamic phenomenon. In this framework, the dynamic pressure,
$P_{dyn}=n_{sw}m_p v_x^2$, serves as the driving force on the system, where $n_{sw}$, $m_p$, and $v_x$ are the number density,
proton mass, and the x component of the solar wind velocity in the GSM coordinates, respectively.
The system's restoring force is described by, $P_b=B^2/2\mu_0$, where $B$ is the total magnetic field at
magnetopause and $\mu_0$ is the vacuum permeability. After accounting for damping and non-ideal effects,
$P_{damp}$, meanwhile neglecting the complex coupling interactions, the momentum equation for the
magnetosheath in a unit cylinder can be represented by equation (1).
$$M_{msh}a_{msh} = P_b - P_{dyn} - P_{damp} \qquad (1)$$
Given that the derivation process of the foundational formula is the same as our previous work,
and this paper is focused on model predictions rather than physical processes, we will refrain from



reiterating it here. The relationship depicting the temporal evolution of the magnetopause position ($r$)
are introduced in equation (2).
$$n_{sw}m_p r\ddot{\boldsymbol{r}} = \frac{(\lambda \boldsymbol{B}_d(r,\theta,\varphi) + \boldsymbol{B}_c(\boldsymbol{r}))^2}{2\mu_0} - n_{sw}m_p \boldsymbol{v}_x^2 \cos^2\alpha - k\Sigma_p \boldsymbol{B}_p^2\dot{\boldsymbol{r}} - \eta\dot{\boldsymbol{r}}/\boldsymbol{r} \qquad (2)$$

Where ($r,\theta,\varphi$) represents the corresponding spherical coordinates, $r$ is the radial distance, $\theta$ is the
latitude angle between [-90°, 90°] and $\varphi$ is the longitude angle adjusted to [-180°, 180°] with 0°
oriented towards the Sun for simplicity, $\boldsymbol{n}$ is the normal direction of magnetopause (Gu et al., 2023).
This simplified equation enables us to capture the fundamental dynamics of the magnetopause's
response to solar wind fluctuations while ensuring computational efficiency. The first term on the right
side of equation (2) signifies the restoring force $\boldsymbol{P}_b$. Here, $\boldsymbol{B}_d$ denotes the Earth's dipole field, $\lambda$ is the
magnetospheric compressibility coefficient, and $\boldsymbol{B}_c$ accounts for contributions from various
magnetospheric currents. The second term on the right side represents the driving force $\boldsymbol{P}_{dyn}$, where $\alpha$
denotes the angle between the x-direction and the normal direction of the magnetopause. The third
term on the right side of equation (2) characterizes a position-dependent dragging effect estimated
from the ionosphere, while the fourth term illustrates a global non-ideal viscous effect. $\boldsymbol{B}_p$ is the
estimated ionospheric magnetic field in the polar region, $\Sigma_p$ stands for the equivalent Pederson
conductivity, $k$ serves as a position-dependent mapping factor, and $\eta$ represents the viscous coefficient.
The final two terms contribute to the damping and non-ideal effects of the system, denoted as $\boldsymbol{P}_{damp}$.
Equation (2) provides a foundation for developing a time-dependent magnetopause model that can
reflect the system's dynamic behaviour more accurately compared to conventional time-independent
models. We will introduce the key parameters in detail in the following sections. All the parameters
are in SI units.
In this study, we incorporate the impact of IMF $\boldsymbol{B}$z and $\Phi$ in the magnetospheric magnetic
pressure. To determine the equation's final fitting coefficients, we performed 1000 independent
iterations of randomly sampling 5000 MCEs sampled randomly from our dataset (a total of 38,887
MCEs).



**3.1 The magnetospheric compressibility coefficient (λ)**
The magnetospheric compressibility coefficient, λ, measures the magnetosphere's response to
solar wind pressure, specifically the ratio of the magnetospheric magnetic field to the pure dipolar
magnetic field (Spreiter et al., 1966; Schield, 1969). This coefficient is one of the most critical
parameters directly affecting the position of the magnetopause. Typically, λ has a value of 2.44 at the
subsolar point, but it changes as the magnetopause shifts and varies with latitude and longitude,
suggesting a more complex formulation (Shue et al., 2011; Chen et al., 2023).  Mead and Beard (1964)
used a self-consistent method, discovered an inward concave structure at the higher latitude
magnetopause, which is influenced by the inclination angle of the Earth's dipole. Their work also
determined the surface shape of the magnetopause when the solar wind flow is perpendicular to the
dipole axis (Φ = 0°), providing an expression for λ as a function of the angles θ and φ.
Several models have been developed to examine the influence of Φ (the angle between Earth's
magnetic axis and the solar wind direction) on the magnetopause's position and shape, offering
valuable insights into the magnetosphere's three-dimensional structure(Formisano et al., 1979;
Boardsen et al., 2000; Lin et al., 2010). These models predict an asymmetric response of the
magnetosphere to variations in Φ. Boardsen et al. (2000) quantified the effects of Φ on the higher
latitude magnetopause using MCEs data from the northern hemisphere. Their work revealed how the
dipole tilt angle influences the magnetopause structure in polar regions, which are particularly sensitive
to changes in the orientation of Earth's magnetic field relative to the solar wind. Lin et al. (2010) further
demonstrated that an increase in Φ causes a slight shift in the centres of the magnetopause cross-
sections, moving them towards the negative Z direction in the subsolar region and towards the positive
Z direction in the tail region. Olson (1969) provided a more detailed representation of λ for various tilt
angles (Φ= 0°, 10°, 20°, 30°) on a 15° by 15° grid of θ and φ values. Building from previous work (Gu
et al., 2023) and considering the influence of Φ on different position of magnetopause λ (θ,φ), we
present a more precise expression of λ tailored to our model, as shown in equation (3):



$$A = \tanh\left[5.568(|\theta| - 0.5325)\right] + 1.0$$
$$\lambda(\theta, \varphi, \Phi) = 2.44 - (0.4 + 0.3A)(\theta + 0.2A\Phi)^2 + (1.0 - 0.5A|\Phi|)\,\varphi^2 \qquad (3)$$

**3.2 Contributions from various magnetospheric currents (Bc)**
Previous studies on the impact of magnetospheric currents on the position of the magnetopause
led to the development of a static magnetopause current model, where the magnetic field of
magnetopause surface current and tail current were fitted using polynomials to reveal the relationship
between variations in the magnetospheric magnetic field and changes in magnetopause position ,e.g.
$\boldsymbol{B}_{surf}(r,\theta,\varphi)$ and $\boldsymbol{B}_{tail}(r,\theta,\varphi)$ (Choe and Beard, 1974b, a; Matsuoka et al., 1995). While our previous
approach, which utilize piecewise functions, may yield discontinuous and non-physical results at the
transition point (Gu et al., 2023). To address this limitation, the basic form of $\boldsymbol{B}_{c0}(r,\theta,\varphi)$ in this paper
is shown below:
$$B_{c0}(r) = [-401904/(\frac{r}{\mathrm{R}_E})^4 + 65489/(\frac{r}{\mathrm{R}_E})^3 + 1500/(\frac{r}{\mathrm{R}_E})^2 - 40][1 + 0.4\sin(2\theta)^2][1.0 - 0.1\sin(\varphi)]\times 10^{-9} \quad (4)$$
In addition, the influence of IMF $\boldsymbol{B}_z$ on the magnetopause position is directional dependent. A
southward IMF may trigger magnetic reconnection at the dayside magnetopause, a significant effect
that is incorporated in most existing models (Aubry et al., 1970; Dungey, 1961; Fairfield, 1971). In
this study, we employ a hyperbolic tangent function, similar to that used in the S98 model, to better
depict the impact of IMF on the magnetopause dynamics. Finally, by considering the impact of IMF
$\boldsymbol{B}_z$ and $\boldsymbol{P}_{dyn}$, and assuming that $\boldsymbol{B}_c$ would behaves differently depending on $\theta$ and $\varphi$, $\boldsymbol{B}_c$ is expressed as
shown in equation (5):
$$B_c = B_{c0}(r,\theta,\varphi)f(B_z)f(P_{dyn})$$
$$= B_{c0}(r)[1.8 + \tanh(-0.3Bz + 6.14)][1 + 0.1P_{dyn}] \qquad (5)$$

This formulation of $\boldsymbol{B}_c$ provides a more refined and physically accurate depiction of the influence
of the magnetospheric current system on magnetopause dynamics. By incorporating the dependence
of $\boldsymbol{B}_c$ on IMF $\boldsymbol{B}_z$ and $\boldsymbol{P}_{dyn}$, and specific locations on the magnetopause, the expression captures the

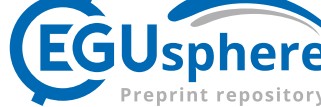

complex spatial variations in the magnetospheric current system that contribute to magnetic pressure,
making our model fully three-dimensional.
**3.3 The damping items**
The damping terms, represented by the last two terms on the right side of the equation, are the
same as our previous work (Chen and Wolf, 1999; Wang and Chen, 2008; Gu et al., 2023). Here,
$B_p$=3×10$^{-5}$ T, represents the estimated ionospheric magnetic field in the polar region, and Σp =3.4S is
the equivalent Pederson conductivity. The viscous coefficient is artificially set as $\eta$=2×10$^{-8}$. The
position-dependent mapping factor $k$ is defined as in equation (6):
$$k = [196(0.05 + e^{-0.05(r/R_E)^2}) - 3.2|\theta| - 1.6|\varphi|] \times 10^{-7} \qquad (6)$$

**4 Result**
By substituting the relevant parameters into equation (2) and assuming the initial shape of the
magnetopause as a paraboloid, x=−0.03(y$^2$+z$^2$) +$R_0$, where $R_0$ is determined by the pressure balance at
the subsolar point, the position of each point on the magnetopause can be computed instantaneously
on a personal computer. The prediction accuracy of the POS model is then evaluated and compared
with other notable time-independent models mentioned earlier. We use the root-mean-square error
(RMSE), denoted as Δ, to quantify the prediction accuracy by comparing the model's calculations with
MCEs observations. A dataset of 38,887 MCEs observed by the THEMIS and CLUSTER satellites is
used for testing. To evaluate the performance of the POS model relative to other models, we calculate
the ratio δ(Δ)/Δ$_{POS}$, where δ(Δ) represents the difference in RMSE between a previous model and the
POS model, and Δ$_{POS}$ is the RMSE of the POS model. This comparison is conducted from various
perspectives. The probability density distributions of RMSE for each model are illustrated in Figure 4.
It can be seen that all models are capable of adequately predicting magnetopause positions, with
the majority (> 70%) showing RMSE within 1 RE. Our model demonstrates superior accuracy, with



80% of its prediction errors falling below 1 $R_E$. Predicting the magnetopause under disturbed solar
wind conditions is more challenging, while the POS model shows improved performance in such
conditions, with 60% of predictions remaining within 1 $R_E$. Given the inherent asymmetry of the
magnetopause, we evaluated the models' performance in both the flank region ($|\varphi| \geq 60°$) and the higher
latitude region ($|\theta| \geq 30°$). The POS model consistently outperforms the others in both regions,
especially in the flank region. Notably, as a time-dependent three-dimensional model, the POS model
seldom produces poor predictions, with RMSE exceeding 3 $R_E$ in only rare cases.

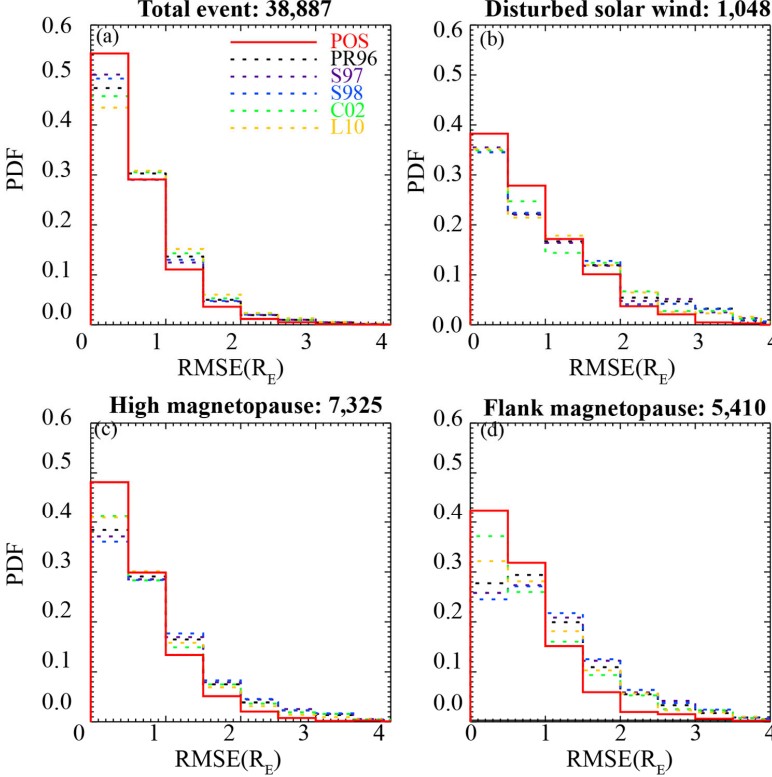


**Figure 4. Distribution of models' RMSE in total(a) and in disturbed solar wind(b); (c) and (d) is the**
**prediction ability in higher latitude magnetopause ($|\theta| \geq 30°$) and in magnetopause flank region ($|\varphi| \geq 60°$)**



**4.1 Time-dependent feature**

The models' prediction accuracy is listed in Table 3, it can be seen that all evaluated models exhibit remarkable predictive capabilities, with $\Delta < 1$ $R_E$, aligning closely with other statistical results found in the literature (Staples et al., 2020b). While it should be noted that the $\Delta$ values calculated for other models in this study may slightly differ from those reported in their original papers. This discrepancy arises due to our use of a significantly larger MCE dataset for comparison.

**Table 3 Models' prediction accuracy for all MCEs and in disturbed solar wind.**

| Model Name | Total (38,887 MCEs) | | $[\sigma(P_{dyn})/< P_{dyn}>] > 100\%$ (1,048 MCEs) | |
|---|---|---|---|---|
| | $\Delta(R_E)$ | $\delta\,(\Delta)\,/\,\Delta_{POS}$ | $\Delta(R_E)$ | $\delta\,(\Delta)\,/\,\Delta_{POS}$ |
| PR96 | 0.899 | +16.9% | 1.389 | +26.4% |
| S97 | 0.884 | +15.0% | 1.383 | +25.8% |
| S98 | 0.894 | +16.3% | 1.388 | +26.3% |
| C02 | 0.926 | +20.4% | 1.325 | +20.6% |
| L10 | 0.960 | +24.8% | 1.377 | +25.3% |
| POS | 0.769 | Average:18.7% | 1.099 | Average:24.9% |

Notably, the POS model demonstrates superior predictive performance, with an average improvement of 18.7% over the other models. Additionally, time-independent models have inherent limitations in capturing the dynamic response of the magnetosphere to solar wind fluctuations, particularly when the magnetopause standoff distance is not in phase with $P_{dyn}$ (Archer et al., 2019). In cases of highly disturbed upstream solar wind, where ratio of standard deviation of $P_{dyn}$ to average $P_{dyn}$ ($\sigma(P_{dyn})/\langle P_{dyn}\rangle$) exceeding 100%, the POS model shows an even greater improvement in predictive accuracy, with a 24.9% enhancement compared to other models. These results suggest that by incorporating time-dependent effects into magnetopause modelling, particularly during periods of solar wind disturbance, the POS model can more effectively capture the non-linear and out-of-phase responses of the magnetopause to rapidly changing solar wind conditions. This results also indicate



that time-dependent model represents an obvious advancement in predicting and understanding
magnetospheric dynamics across a wide range of solar wind conditions.

The magnetopause is rarely static, exhibiting continuous motion under varying solar wind
conditions and displaying complex dynamics during both intense disturbances and gentle changes. A
notable feature of these dynamics is the periodic oscillation within the Pc5 frequency range (2-7 mHz),
often termed the "magic frequency" in magnetospheric physics (Samson et al., 1992; Plaschke et al.,
2009a; Plaschke et al., 2009b). The magnetopause oscillations can be driven by quasi-periodic solar
wind dynamic fluctuations, or explained by magnetospheric cavity mode and Kruskal-Schwarzschild
mode (Archer et al., 2013; Kruskal and Schwarzschild, 1954; Kepko and Spence, 2003; Kivelson et
al., 1984). Our previous research indicates that the oscillations of the magnetopause ought to have
eigenfrequencies ($f_0$) which are determined by the restoring force ($P_B$), the external driving force ($P_{dyn}$)
as well as the damping force ($P_{damp}$) (David Halliday, 2021; Freeman et al., 1995; Gu et al.,
2023). Magnetopause will responses to solar wind with phase difference ranging from 0 to 180 degrees,
depending on the driving frequency of the solar wind ($f_{drive}$). The magnetopause behaves as a low-pass
filter, effectively screening out very high-frequency solar wind fluctuations (e.g., $f_{drive}>15f_0$, where $f_0$
is the eigenfrequency of the magnetopause). This filtering effect results in smoother predictions of
magnetopause behaviour which could be found in Figure 5. For relatively high fluctuations (e.g., $15f_0>$
$f_{drive} >2f_0$), the phase difference between the solar wind and magnetopause approaches 180 degrees,
indicating an anti-phase response. At resonance ($f_{drive}≈f_0$), the magnetopause exhibits a 90-degree
phase lag relative to the solar wind forcing. Conversely, the magnetopause only behaves in-phase with
the solar wind under low-frequency fluctuations ($f_{drive}<0.5f_0$), which is the scenario typically revealed
by time-independent models.



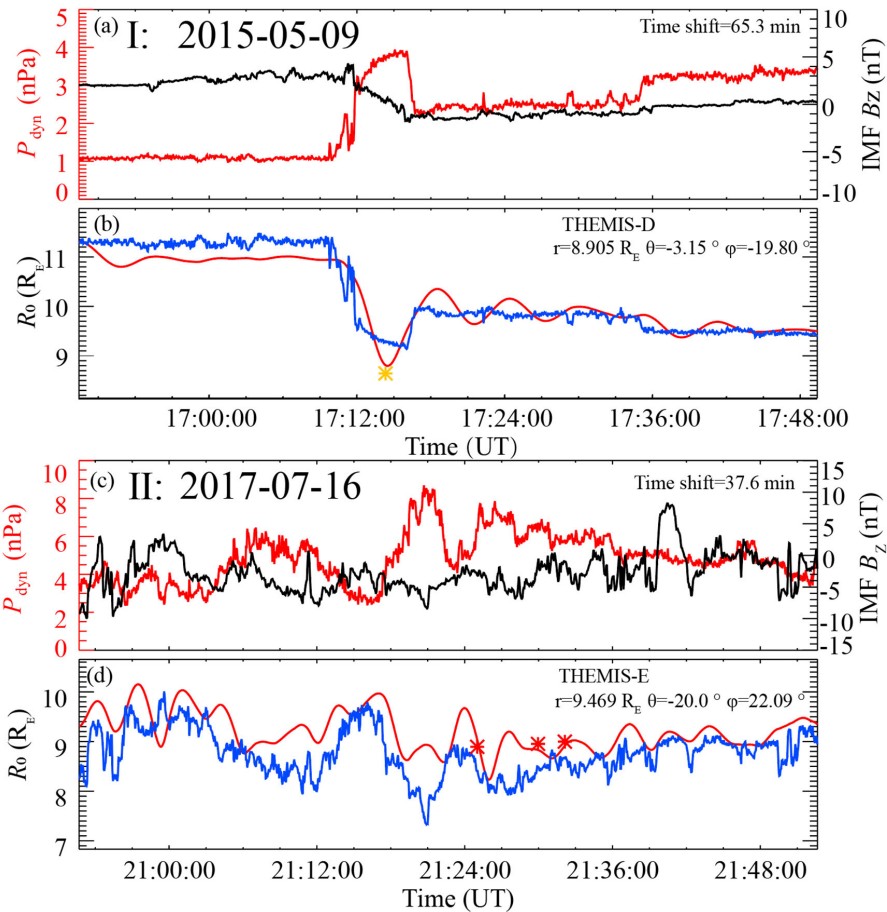


**Figure 5. Case study for the overall oscillation of magnetopause using time-independent model S98 and time-dependent POS model to predict the its position. (a), (c) The corresponding upstream solar wind dynamic pressure $P_{dyn}$ (red line) and interplanetary magnetic field $B_z$ component (black line) observed by Wind with a time shift of 65.3 min and 37.6 min, respectively; (b), (d)The predictions of S98(blue) and POS (red) model's prediction based the input solar wind and the subsolar point position projected from THEMIS. The asterisks represent the positions of MCEs observed by THEMIS.**

The time-dependent POS model demonstrates the capability to depict these magnetosphere oscillations and the phase difference accurately. Figure 5 presents two specific cases illustrating the POS model's performance. In both cases, the model predicts quasi-periodic oscillations in the magnetopause that align well with consecutive THEMIS MCE observations. In Case I, both models

- 18 -





initially predict the magnetopause position at ~11.5 $R_E$ before a pressure pulse in solar wind. The POS
model uniquely predicts four oscillations around its equilibrium position (~10 $R_E$) before the
magnetopause reaches a new pressure balance. This dynamic behaviour cannot be physically captured
by any time-independent models. In Case II, the POS model accurately captures the oscillations around
21:24 UT-21:33 UT, which are not all in-phase with the solar wind dynamic pressure ($P_{dyn}$). Notably,
the POS model depicts anti-phase responses observed in the second and third crossings, while the S98
model shows a reverse trend in motion that deviates more from observations. These results suggest
that by incorporating time-dependent effects into magnetopause modelling, particularly during periods
of solar wind disturbance, the POS model can more effectively capture the non-linear and out-of-phase
responses of the magnetopause to rapidly changing solar wind conditions.
**4.2 Three-dimensional characteristic**

The POS model developed here incorporates the asymmetrical effects of dipole tilt angles,

latitude, and longitude differences, as integrated into equations (2) and (3). The model's parameters
were comprehensively calibrated, allowing it to more accurately depict the three-dimensional shape of
the magnetopause. To assess its validity across different magnetopause regions, extensive tests were
performed, with results presented in Table 4. In the higher latitude magnetopause ($|\theta| \geq 30°$), a region
where many models face challenges, the POS model, alongside the L10 model, demonstrates superior
performance, showing an impressive 28.7% improvement in accuracy compared to other models.
Similarly, in the flank regions ($|\varphi| \geq 60°$), where surface waves and other magnetospheric fluctuations
complicate position and shape determination, the POS model maintains its high accuracy, with a 35.2%
improvement over other models. These results suggest that the POS model offers a more accurate and
comprehensive representation of the magnetopause across its entire structure, outperforming other
models in both higher latitude and flank regions.






**Table 4  Models' prediction accuracy for higher latitude and flank regions**

| Model name | $\|\theta\| \geq 30\,°$ (7,325 MCEs) | | $\|\varphi\| \geq 60\,°$ (5,410 MCEs) | |
|---|---|---|---|---|
| | $<\Delta>(R_E)$ | $\delta\,(\Delta)/\Delta_{POS}$ | $\Delta(R_E)$ | $\delta\,(\Delta)/\Delta_{POS}$ |
| PR96 | 1.149 | +29.5% | 1.315 | +33.6% |
| S97 | 1.180 | +33.0% | 1.388 | +41.1% |
| S98 | 1.195 | +34.7% | 1.403 | +42.6% |
| C02 | 1.130 | +27.4% | 1.268 | +28.9% |
| L10 | 1.053 | +18.7% | 1.278 | +29.9% |
| POS | 0.887 | Average:28.7% | 0.984 | Average:35.2% |


Surface waves are a distinct feature of the magnetopause, originating from various factors
including solar wind and bow shock dynamics, as well as instabilities within the magnetopause and
magnetosphere under specific conditions. Several localized physical processes have been identified as
potential drivers of these surface waves, including the Kelvin-Helmholtz instability, magnetic
reconnection and flux transfer event (Hartinger et al., 2013; Agapitov et al., 2009; Archer et al., 2021).
It is also found tailward-moving surface wavelet could be driven by disturbed solar wind (large σ
($P_{dyn}$)/< $P_{dyn}$> ) (Sibeck et al., 1989). Our previous study has revealed a distinct mechanism for the
formation of surface wave-like structures in the magnetopause (Gu et al., 2023). The interplay between
dynamic pressure ($P_{dyn}$), magnetic pressure ($P_b$), and damping pressure ($P_{damp}$) results in different
oscillation periods at various points on the magnetopause. These variations create a time lag within the
magnetopause structure, manifesting as a surface wave-like pattern. Figure 6 shows a surface wave-
like structure predicted by the POS model during relatively disturbed upstream solar wind conditions.
The POS model's predictions are compared with those of the C02 model, which has been demonstrated
as the most effective time-independent model in the flank region according to our evaluation. Figure
6 (a) displays the solar wind dynamic pressure and the north-south component of the interplanetary
magnetic field. The radial positions of various points on the magnetopause in the XY plane (Z=0), as



calculated by the POS model, are traced in Figure 6(b). Notably, the magnetopause shapes calculated
in Figures 6(c)-(e) reveal surface wave-like structures evolving over time. THEMIS MCEs observed
in the flank region corroborate this predicted surface wave-like structure, indicating that the
magnetopause position predicted by the POS model is more accurate than that predicted by the C02
model.

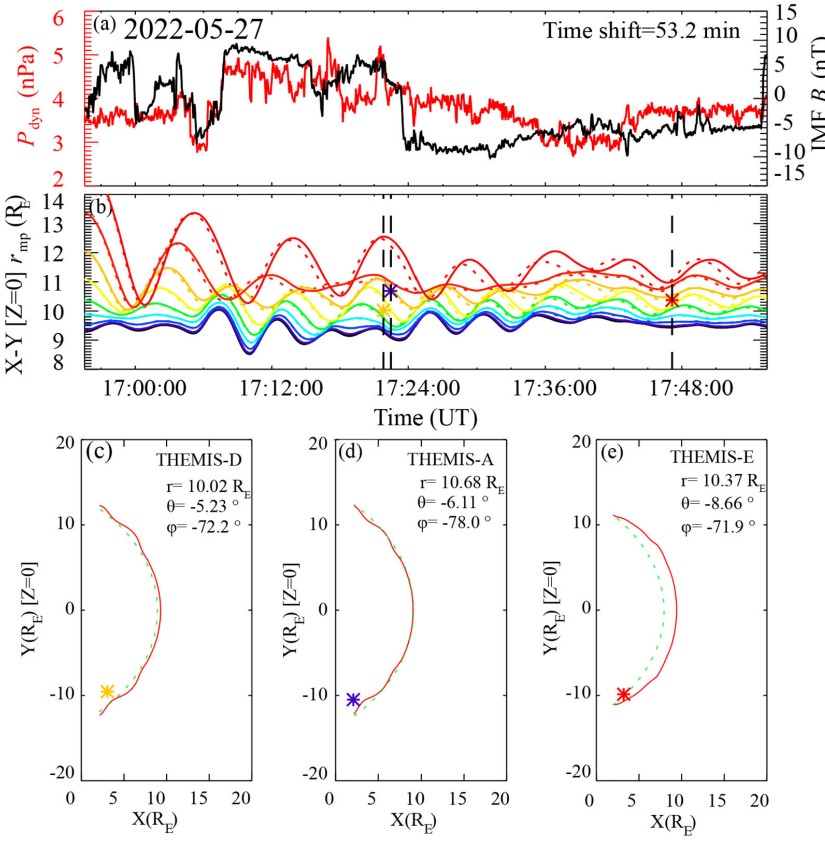


**Figure 6. A surface wave-like structure in X-Y magnetopause flank region. (a) The corresponding solar wind**
**dynamic pressure(red) and IMF $B_z$ component(black), (b) The red, orange, yellow, green, blue, purple and**
**black colours represent the initial magnetopause positions at φ =±80°, ±70°, ±60°, ±50°, ±40°, ±30°,**
**±20°, ±10°, 0° , respectively (dot line is the corresponding negative value of φ). The asterisk in purple**
**(THEMIS-A), yellow (THEMIS-D) and red (THEMIS-E) indicate the satellite observation of MCEs**




**projected onto the X-Y(Z=0) plane; (c) (d) (e) The shape of magnetopause in the X-Y plane at different time**
**predicted by POS model (red dash line) and C02 model (green dot line).**
The POS model's predictions are compared with those of the C02 model, which has been
demonstrated as the most effective time-independent model in the flank region according to our
evaluation. Figure 6 illustrates a surface wave-like structure predicted by the POS model during
relatively disturbed upstream solar wind conditions. The predicted surface wave-like structure is
corroborated by THEMIS MCEs in the flank region, where the actual magnetopause position is closer
to Earth than predicted by C02 model.
**5 Discussion and Conclusion**
Accurately calculating the position of the magnetopause is essential for space weather forecasting
and understanding the underlying physical mechanisms involved in the solar wind- magnetosphere
interaction. In this work, we developed the POS model, the first time-dependent three-dimensional
magnetopause model based on quasi-elastodynamic theory. By incorporating key solar wind
parameters such as $P_{dyn}$, IMF $B_z$ and $\Phi$, this model effectively depicts magnetopause dynamics. The
POS model offers a new approach to describing magnetopause position, overall oscillation, and surface
wave-like structures as interconnected phenomena. Its time-dependent feature excels in capturing
dynamic processes, particularly under highly disturbed solar wind conditions. The three-dimensional
nature allows for accurate depiction of the overall magnetopause shape, with notable precision in
higher latitude regions and flank areas. This capability addresses limitations in existing models and
provides a more comprehensive picture of magnetopause dynamics from a different perspective.
However, there are still limitations and areas for improvement that future research should address:
(1) Adapting to extreme solar wind conditions: Similar to the force-deformation relationship of a
spring that requires a specific range of applicability, the POS model has not been specifically optimized
for extreme solar wind conditions (e.g., $P_{dyn}$<0.5 nPa and $P_{dyn}$>10 nPa). When the solar wind dynamic
pressure is low, the quasi-elastic process between the solar wind and the magnetopause exhibits



stronger damping characteristics, while at very high solar wind dynamic pressures, the magnetopause
shows increased rigidity. Future iterations could incorporate more suitable damping coefficients and
include $P_{dyn}$ in the magnetospheric compressibility coefficient to broaden the model's applicability
range.
(2) Incorporating additional solar wind factors: Existing research has shown that even under
similar solar wind dynamic pressure conditions, changes in solar wind density and velocity have
distinct effects on magnetopause position (Samsonov et al., 2020). Additionally, the influence of solar
wind temperature, more comprehensive IMF effects (e.g., $B_x$ and $B_y$), and other solar wind components
(e.g., alpha particles) on magnetopause position are not reflected in the current model. Future models
could consider introducing these factors to achieve better predictive results.
(3) Better nightside extension: The current POS model is primarily based on the dayside quasi-
elastodynamic theory and is calibrated and validated using dayside MCEs. In the future, the model's
calculation results for the nightside region could be improved by combining the fitting approach of
empirical models with a more flexible curve function calibrated using a larger number of nightside
MCE observations.
(4) Cusp region representation: Accurately modelling the magnetopause cusp region, shaped by
Earth's dipole field, remains challenging. While some models approximate this region by fitting two
distinct curves, capturing its shape and position precisely is complex. Improving the representation of
the cusp region will require further analysis of higher latitude satellite data to enhance model accuracy.
(5) Parameter fine-tuning: Further refinement of model parameters, potentially through machine
learning techniques or implementing piecewise functions for different regions, could improve the
model's accuracy. However, as noted in the introduction, it's important to balance model complexity
with practicality. Overly complex parameter expressions can lead to increased inconvenience and



higher computational costs. For those seeking the highest possible prediction accuracy, a more
practical approach might involve using numerical simulations.
The upcoming SMILE mission (Solar wind Magnetosphere Ionosphere Link Explorer), a joint
mission between the Chinese Academy of Sciences and the European Space Agency, is set to launch
in 2025. This mission will provide more detailed data on magnetopause position and polar cap shape
over time, enhancing the ability to validate and refine existing magnetopause models.
In summary, this study introduces the POS model, the first time-dependent three-dimensional
magnetopause model based on quasi-elastodynamic theory. Unlike time-independent models, the POS
model effectively captures the dynamic movement of the magnetopause under varying solar wind
conditions. When compared to five widely used models, the POS model demonstrates superior
predictive accuracy, showing a 18.7% improvement with RMSE=0.768 $R_E$. As a time-dependent
model, it demonstrated superior accuracy under highly disturbed solar wind conditions (24.9% better).
Its three-dimensional nature allows for enhanced accuracy in higher latitude regions (28.7% better)
and flank regions (35.2% better) of the magnetopause. Moreover, compared to numerical simulations,
the POS model offers a concise formulation with rapid computational speed, making it feasible for
direct deployment on satellites in the future, where onboard chips could complete calculations, greatly
enhancing satellite intelligence. By providing a more precise and dynamic representation of the
magnetopause, the POS model enhances our ability to predict and analyse space weather events and
may also offer new insights and methodologies for developing magnetopause models for other planets.

**Code and data availability**
The current version of model is available from the project
website: http://www.spaceweather.org.cn/pos_model. The exact version of the model used to produce
the results used in this paper is archived on Zenodo: https://doi.org/10.5281/zenodo.14189153 . The



URL includes the code and the list of MCEs used in this paper. The data of THEMIS satellite can be
obtained from  https://cdaweb.gsfc.nasa.gov/pub/data/themis/ , and the data of WIND satellite can be
obtained from https://cdaweb.gsfc.nasa.gov/pub/data/wind/ .
**Competing interests**
The contact author has declared that none of the authors has any competing interests.
**Author contribution**
Y. W. designed the model. Y.X.G developed the model code and carried them out. Y. X. G and
Y. W. prepared the original manuscript. Y. X. G and X. J. S. prepared the MCE list. F. S. W., X. S. F.,
A. S., X. J. S., B. Y. W., P. B. Z., C. W. J., Y. L. C, X. J. X. and Z. L. Z. discussed the scientific results,
reviewed and revised the manuscript.
**Acknowledgements**
We thank NASA/GSFC CDAWeb for providing the Wind, THEMIS and Cluster data. This work
is jointly supported by the National Natural Science Foundation of China 42174199, China Scholarship
Council   202306120305,   Guangdong   Basic   and   Applied   Basic   Research   Foundation
2023B1515040021,   and   Shenzhen   Technology   Project   JCYJ20210324121210027   and
RCJC20210609104422048,   and   Shenzhen   Key   Laboratory   Launching   Project   No.
ZDSYS20210702140800001.



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
