# Peer review of "1A Time-Dependent Three-Dimensional Magnetopause Model2Based on Quasi-elastodynamic Theory"

_EGUsphere, 2024_

## Referee Comment (RC2)

The paper introduces a time-dependent three-dimensional magnetopause model, named the POS (Position-Oscillation-Surface wave) model, based on quasi-elastodynamic theory, addressing limitations of existing time-independent models. It emphasizes the dynamic behavior of the magnetopause, which includes oscillations and surface wave-like structures in response to solar wind variations. The model was validated against 38,887 observed magnetopause crossing events, achieving a root-mean-square error of 0.768 Earth radii, marking an 18.7% improvement over five widely used models. Notably, the POS model showed enhanced accuracy under disturbed solar wind conditions and in specific magnetopause regions. The study highlights the need for new strategies to balance dynamic representation and computational feasibility in magnetopause modeling. The paper is well written, and I have few concerns to be addressed before the paper will be acceptable for publication.

Lines 61 – 69: I would suggest the authors to include the results presented by Collado-Vega et al., (2023) in this discussion. They compared the magnetopause predictions obtained by different MHD models, showing the discrepancies for the standoff positions.
**https://doi.org/10.1029/2022SW003212**

Lines 147 – 150: Are the final dataset (38,887) single magnetopause crossings? When all five THEMIS/Cluster satellite cross the magnetopause in a short time interval, it is counted as one or multiple crossings? What is the maximum time interval between two consecutive MCEs to be considered redundant? This information is not clear in the text.

Line 153: Please clarify the 300 s condition (why 300?).

Line 153 – 155: The one-hour average velocity is based on which assumption? Is it reasonable to use the same 1h average for all solar cycle phase? How different/better is this methodology from the time shift provided by OMNI service? How much better is Wind data compared to ACE or other available solar wind monitors?

Line 222: There is no **n** (normal) indicated in the equation 2. Please, check.

Line 255: typo "... three-dimensional structure (Formisano et al., 1979; ...".

Line 292: "two terms on the right side of the equation ...".  Suggest labelling the equation.

Line 314 – 317: Have the authors analyzed the results for the flank regions independently? Is there any asymmetry in the results?

Lines 363 – 382: Figure and the following discussion. Are the authors projecting the THEMIS position at the subsolar region? In this case would not be more interesting to calculate the magnetopause model results at the real THEMIS location? On my understanding, when the authors analyze the projection of the spacecraft position at the subsolar point, they are

assuming the magnetopause is "shrinking" and could be neglecting any wave motion along the surface caused by a pressure pulse. Please, make a comment on that point.

Lines 422 – 434: Figure 6 and the following discussion.  The authors claim that the POS model's predictions are more effective than C02 model. However, the results on Figure 6b are considering $\theta = 0$ and the spacecraft locations are not. Also, in comparison with the satellite positions both models show very close results (by visual analysis) on figure 6C, the POS is closer on figure 6d and C02 is closer in figure 6e. I suggest the authors to point out these discrepancies.

Discussion and Conclusion

How does the POS model respond to transient events formed at the bow shock or in the magnetosheath when they reach the magnetosphere? See Sibeck et al., (2022) https://doi.org/10.1029/2022JA030704 and Silveira & Sibeck (2023) https://doi.org/10.1029/2023JA031362 .

Is POS model capable to catch return magnetic fluxes from nightside to dayside due to nightside magnetic reconnection?  See Silveira et al., (2024) **https://doi.org/10.1029/2023JA032166**.

---

## Author Comment (AC2)

Dear Editor, Dear Reviewers,

Thank you very much for reviewing our manuscript and providing such professional and detailed comments. We greatly appreciate the time and effort devoted to evaluating our work and offering valuable suggestions, which have significantly helped us improve the quality of our manuscript.

We acknowledge and adhere to the journal's current submission protocol, which requires the initial submission of our responses to the reviewers' comments prior to the upload of the revised manuscript. Accordingly, we have provided our detailed, point-by-point responses below. We will promptly upload the revised manuscript upon receiving confirmation that we may proceed, or as otherwise instructed by the journal's guidelines.

Sincerely,

Yi Wang
* * *
Reviewer 1
* * *
**The dayside magnetopause constitutes a critical interface for the injection of energy and matter from the solar wind into Earth's magnetosphere. Establishing the spatial configuration of the dayside magnetopause is therefore of paramount importance. The manuscript under review introduces a time-dependent three-dimensional dayside magnetopause model which has been shown to offer superior accuracy over existing static models. I recommend this manuscript for publication following some minor revisions.**

**1. Title Specificity: Given that the model's applicability is confined to the dayside region, it would be more precise to amend the title to "A Time-Dependent Three-Dimensional Dayside Magnetopause Model". This change will better reflect the scope and focus of the research presented.**
R: The title has been revised to "A Time-Dependent Three-Dimensional **Dayside** Magnetopause Model Based on Quasi-Elastodynamic Theory" as you suggested.

**2. Clarification of Mathematical Derivations: The proposed magnetopause model incorporates several complex equations (equations 3-6). However, the authors have not provided sufficient explanation regarding the derivation of these expressions. For the benefit of readers who may wish to understand or further develop the model, it would be valuable to include a more detailed exposition of how these equations were formulated. This could take the form of additional text or figures within the manuscript.**

R: We have added the following changes:

**Line 268-274 (for equation 3):** According to Olson (1969), who provided a detailed representation of λ for various tilt angles ($\Phi$ = 0°, 10°, 20°, 30°) on a 15° × 15° grid of θ and φ values, the dipole tilt angle significantly alters the fundamental behaviour of λ, with varying effects on θ and φ. This influence is more pronounced in higher latitude regions (θ > 30°). Building on previous work by Gu, et al. (2023) , in which $\lambda = 2.44 - 0.4\theta^2 + \varphi^2$ , and incorporating the effect of $\Phi$ at different positions of the magnetopause λ(θ, φ), we derive a more precise expression for λ, specifically tailored to our model, as presented in Equation (3):

**Line 281-287 (for equation 4) :** In our earlier study (Gu, et al. 2023), the magnetic field of the current system, denoted as $B_c(r)$, did not account for variations in θ and φ. This limitation is addressed in the present study. The fundamental form of $B_{c0}(r,\theta,\phi)$ , incorporating these angular dependencies, is introduced in Equation (4). The current system exhibits asymmetry effects, consistent with magnetospheric magnetic field models such as T96 and T01(Tsyganenko 2001; Tsyganenko 1996). These models incorporate dawn-dusk asymmetry in the magnetospheric current, reflecting the influence of these angular dependencies.

**Line 289-296 (for equation 5):** In our previous work, $\boldsymbol{B}_c(r)$ was defined as a piecewise function of $\boldsymbol{P}_{dyn,}$ which could yield discontinuous and non-physical results at the transition points (Gu, et al. 2023). To address this limitation, we now consider the impact of $\boldsymbol{P}_{dyn}$ in a continuous form, eliminating the piecewise dependence. Furthermore, the impact of the IMF $\boldsymbol{B}_z$ on the magnetopause position is directionally dependent, with a southward IMF triggering dayside magnetic reconnection—an essential process already incorporated in most existing models (Aubry, et al. 1970; Dungey 1961; Fairfield 1971). To quantify this effect, we adopt a hyperbolic tangent function, similar to that in Shue, et al. (1998). Finally, by considering the combined effects of both IMF $\boldsymbol{B}_z$ and $\boldsymbol{P}_{dyn}$, $\boldsymbol{B}_c$ is expressed as in Equation (5):

**Line 304-311 (for equation 6):** The damping terms in our model consist of a position-dependent dragging effect from the ionosphere $\boldsymbol{F}_d = k\Sigma_p \boldsymbol{B}_p^2 \dot{r}_{mp}$ and a global non-ideal viscous effect $\boldsymbol{F}_N = \eta\dot{r}_{mp}/r_{mp}$, consistent with our previous work(Chen and Wolf 1999; Gu, et al. 2023; Wang and Chen 2008). We set $\boldsymbol{B}_p$=3×10$^{-5}$ T to represent the approximate ionospheric magnetic field in the polar region, while $\Sigma_p$ =3.4 S serves as the equivalent Pedersen conductivity. The viscous coefficient is artificially set to $\eta$ =2×10$^{-8}$. As defined in Equation (6), the position-dependent mapping factor $k$ is empirically calibrated based on the magnetopause location (r,θ,φ), increasing when the magnetopause compresses and decreasing with increasing latitude and longitude.

Aubry, M. P., C. T. Russell and M. G. J. J. o. G. R. Kivelson (1970). "Inward motion of the magnetopause before a substorm." **75**: 7018-7031.

Chen, C. X. and R. A. Wolf (1999). "Theory of thin-filament motion in Earth's magnetotail and its application to bursty bulk flows." Journal of Geophysical Research-Space Physics **104**(A7): 14613-14626.

Dungey, J. W. (1961). "Interplanetary Magnetic Field and the Auroral Zones." Physical Review Letters **6**(2): 47-48.

Fairfield, D. H. (1971). "Average and unusual locations of the Earth's magnetopause and bow shock." **76**(28): 6700-6716.

Gu, Y. X., Y. Wang, F. S. Wei, X. S. Feng, X. J. Song, B. Y. Wang, P. B. Zuo, C. W. Jiang, X. J. Xu and Z. L. Zhou (2023). "Quasi-elastodynamic Processes Involved in the Interaction between Solar Wind and Magnetosphere." The Astrophysical Journal **946**(2): 102.

Olson, W. P. (1969). "The shape of the tilted magnetopause." **74**(24): 5642-5651.

Shue, J.-H., P. Song, C. T. Russell, J. T. Steinberg, J. K. Chao, G. Zastenker, O. L. Vaisberg, S. Kokubun, H. J. Singer, T. R. Detman and H. Kawano (1998). "Magnetopause location under extreme solar wind conditions." **103**(A8): 17691-17700.

Wang, Y. and C. X. Chen (2008). "Numerical Simulation of Radial Plasma Transport in the Saturn's Magnetosphere." Chinese Journal of Geophysics **51**: 635-642.

Reviewer 2

**The paper introduces a time-dependent three-dimensional magnetopause model, named the POS (Position-Oscillation-Surface wave) model, based on quasi-elastodynamic theory, addressing limitations of existing time-independent models. It emphasizes the dynamic behavior of the magnetopause, which includes oscillations and surface wave-like structures in response to solar wind variations. The model was validated against 38,887 observed magnetopause crossing events, achieving a root-mean-square error of 0.768 Earth radii, marking an 18.7% improvement over five widely used models. Notably, the POS model showed enhanced accuracy under disturbed solar wind conditions and in specific magnetopause regions. The study highlights the need for new strategies to balance dynamic representation and computational feasibility in magnetopause modeling. The paper is well written, and I have few concerns to be addressed before the paper will be acceptable for publication.**

**Lines 61 – 69: I would suggest the authors to include the results presented by Collado-Vega et al., (2023) in this discussion. They compared the magnetopause predictions obtained by different MHD models, showing the discrepancies for**

the standoff positions. https://doi.org/10.1029/2022SW003212

R: As suggested, we have added the references and expanded the Discussion and Conclusion sections to include the MHD-related aspects.

Please see Lines 71 – 74: *Collado-Vega et al.* (2023) compared the magnetopause predictions obtained by different MHD models, showing the discrepancies for the standoff position. Their analysis also specifically addressed the impact of extreme solar wind conditions, which are known to cause space weather hazards, on the magnetopause.

**Lines 147 – 150: Are the final dataset (38,887) single magnetopause crossings? When all five THEMIS/Cluster satellite cross the magnetopause in a short time interval, it is counted as one or multiple crossings? What is the maximum time interval between two consecutive MCEs to be considered redundant? This information is not clear in the text.**

R: Yes, they are treated as single MCEs in our calculations. Initially, we collected a dataset of 89,911 MCEs, many of which were duplicates. To rigorously validate the predictive capability of our model, we excluded simultaneous crossings on the same satellite. Specifically, two consecutive MCEs occurring within 3 seconds (the resolution of WIND/3DP) are considered redundant. Therefore, if the five THEMIS/Cluster satellites cross the magnetopause within a short time interval (if it exceeds 3 seconds) they are counted as multiple crossings.

To avoid potential misunderstandings, we have revised the text to clarify this point.

Please see Lines 151 – 154: After excluding redundant crossings (i.e., those occurring simultaneously on the same satellite), invalid data (i.e., crossings without valid upstream solar wind observations), and nightside MCEs (where $X_{GSM} < 0$ $R_E$), a total of 38,018 THEMIS MCEs and 869 CLUSTER MCEs (see Figure 2)are selected for this study.

**Line 153: Please clarify the 300 s condition (why 300?).**

[Figure]

R: Since solar wind observations at L1 do not always directly reflect conditions in the near-Earth environment, and real-time measurements closer to Earth are limited, it is necessary to establish a reasonable threshold for matching upstream data. As illustrated, the approximate five-minute (300s) transit time from the bow shock to the

magnetopause (Plaschke, et al. 2013) provides a practical guideline for this threshold. The 300s threshold adopted here strikes a balance between maintaining data quality, capturing a sufficient number of relevant MCEs, and ensuring computational efficiency.

In our work, we segmented the solar wind data from the L1 point into 1-hour intervals and applied a sliding average method to match the upstream data with observations in front of the magnetopause. A 300s potential error window was set for the time shift from L1 to the magnetopause, consistent with Plaschke, et al. (2013). Accurately time-shifting these events is often challenging due to the complexities of solar wind interactions and uncertainties in propagation models. Given the large size of our MCE dataset, limiting the analysis to propagation times below 300s allows us to focus on events with a clearer and more reliable connection between the upstream solar wind and the observed MCEs.

To avoid confusion, we have added the following clarification:
Please see Line 156-157: The 300s threshold is set as the potential error window for the time shift from L1 to the magnetopause.

**Line 153 – 155: The one-hour average velocity is based on which assumption? Is it reasonable to use the same 1h average for all solar cycle phase? How different/better is this methodology from the time shift provided by OMNI service? How much better is Wind data compared to ACE or other available solar wind monitors?**

R: The one-hour average velocity is based on the typical propagation time of the solar wind from L1 to the magnetopause. In our study, we use a one-hour observation window and a 3-second iteration step to derive the better numerical predictions, while parameters can be adjustable (e.g., a 30-minute or 2-hour window, and an iteration step ranging from 3 seconds to 1 minute). The one-hour window serves as a sliding average that fits well for both solar maximum and solar minimum conditions and it will be further calibrated using a maximum acceptable mismatch of 300 seconds.

Accurately determining the solar wind's transport time from L1 to the magnetopause ($\delta t$) is critical for analyzing magnetospheric responses. Shue, et al. (1997) employed a constant time shift, while Chao, et al. (2002) assumed uniform solar wind velocity and calculated $\delta t$ as $S/<v>$. Lin, et al. (2010) further refined this approach by matching magnetic field and plasma data from L1 to magnetosheath satellite observations. Given the difficulty of obtaining joint observations in the near-Earth region as meticulously as Lin, et al. (2010) and the large size of our MCE dataset, we adopted a two-step approach in our work (Gu, et al. 2023). First, we determined $\delta t$ using the method of Chao, et al. (2002), then refined it by applying a sliding velocity method over a 1-hour window, with a maximum acceptable mismatch of 300 seconds.

OMNI data can directly provide the time shift. However, compared to Wind, the OMNI dataset suffers from more frequent data gaps and a lower temporal resolution. This limits its usefulness for our time-varying model, which benefits from a continuous, high-quality data stream and high temporal resolution.

Both WIND and ACE provide steady, long-term measurements, but WIND is particularly favored due to its superior data quality (e.g., less data gaps). Additionally, WIND's higher resolution (3 seconds for plasma and 0.092 seconds for magnetic field) makes it more suitable for capturing dynamic magnetopause changes compared to ACE's coarser resolution (64 seconds for plasma and 16 seconds for magnetic field). A key distinction between time-dependent and time-independent models lies in their treatment of solar wind-magnetopause interactions. Time-independent models establish a direct point-to-point relationship between solar wind conditions and magnetopause responses, while time-dependent models capture the time-varying nature of this relationship. High temporal resolution is important for discerning the time-dependent aspects of magnetopause dynamics. The continuous, high-resolution data from WIND enable our model to discern time-dependent features of magnetopause dynamics.

Additionally, we provide a table below to show the models' prediction accuracy by using OMNI data. You can see that although the lower time resolution of OMNI results in a slight decrease in prediction accuracy compared to WIND, the POS model remains superior to the other models.

Table Models' prediction accuracy for higher latitude and flank regions by using OMNI data

| Model name | $|\theta| \geq 30\,°$ (7,320 MCEs) | | $|\varphi| \geq 60\,°$ (5,321 MCEs) | |
|---|---|---|---|---|
| | $<\Delta>(R_E)$ | $\delta\,(\Delta)/\Delta_{POS}$ | $\Delta(R_E)$ | $\delta\,(\Delta)/\Delta_{POS}$ |
| PR96 | 1.381 | +18.6% | 1.601 | +24.7% |
| S97 | 1.402 | +20.4% | 1.654 | +28.8% |
| S98 | 1.416 | +21.6% | 1.669 | +30.0% |
| C02 | 1.299 | +11.6% | 1.488 | +15.9% |
| L10 | 1.335 | +14.7% | 1.602 | +24.8% |
| POS | 1.164 | Average:17.4% | 1.284 | Average:24.8% |

Based on the reasons above, we have also implemented the following changes:
Line 146-147: The WIND spacecraft, launched into orbit around Earth in 1994 and relocated to Lagrange L1 point after 2004, provides continuous, high-quality in-situ solar wind observations.
Line 154-160: The time shift ($\delta t$) between WIND to the satellite MCE is determined by comparing the time of each crossing (t1) with the probable arrival time of corresponding solar wind observation from WIND (t0 + $\delta t$), satisfying (t0 + $\delta t$)– t1 < 300 s. The 300s

threshold is set as the potential error window for the time shift from L1 to the magnetopause. δt is calculated as (L1-r)/ <vx>, L1 (L1=235 RE) is the distance from the Earth to the L1 point, r denotes the radial position of the magnetopause, and <vx> is the 1-hour sliding average of the solar wind velocity in the x-component (Chao et al., 2002).

**Line 222: There is no n (normal) indicated in the equation 2.**

R: Thank you for pointing out this error. It was indeed a redundant description. We have deleted the unnecessary content " $n$ is the normal direction of magnetopause ", as the spherical coordinates (r, θ, φ) are already clearly defined.

**Please, check. Line 255: typo "… three-dimensional structure (Formisano et al., 1979; …".**

R: We have revised the descriptions into:
Line 256-259:   "Several models (*Boardsen et al.*, 2000; *Formisano et al.*, 1979; *Lin et al.*, 2010) have been developed to investigate the influence of Φ (the angle between Earth's magnetic axis and the solar wind direction) on the magnetopause's position and shape, offering valuable insights into the magnetosphere's three-dimensional structure."

**Line 292: "two terms on the right side of the equation …".  Suggest labelling the equation.**
We have labelled the terms.
Line 302-309: The damping terms in our model consist of a position-dependent dragging effect from the ionosphere $F_d = k\Sigma_p B_p^2 \dot{r}_{mp}$ and a global non-ideal viscous effect $F_N = \eta \dot{r}_{mp}/r_{mp}$, consistent with our previous work(Chen and Wolf 1999; Gu, et al. 2023; Wang and Chen 2008). We set $B_p$=3×10⁻⁵ T to represent the approximate ionospheric magnetic field in the polar region, while Σp =3.4 S serves as the equivalent Pedersen conductivity. The viscous coefficient is artificially set to $\eta$ =2×10⁻⁸. As defined in Equation (6), the position-dependent mapping factor $k$ is empirically calibrated based on the magnetopause location (r,θ,φ), increasing when the magnetopause compresses and decreasing with increasing latitude and longitude.

**Line 314 – 317: Have the authors analyzed the results for the flank regions independently? Is there any asymmetry in the results?**
R: We have examined the flank regions independently, and Figure 6(b) clearly reveals

asymmetry in their responses. Although our primary focus was not on emphasizing this asymmetry, the figure shows that the dawn and dusk flanks respond differently to solar wind variations. The dashed and dotted lines, representing $\pm\varphi$, illustrate these distinct responses.

Among the empirical models we compared, only Lin, et al. (2010) and our model incorporate three-dimensional asymmetry. Lin, et al. (2010) assume that $\varphi$ influences the flare angle of the magnetopause, while in our work, asymmetry is incorporated through $\mathbf{B}_{c0}(r,\theta,\phi)$, as shown in Equation (4). This basic form, representing the current system, is derived from (Choe and Beard 1974a; Choe and Beard 1974b; Matsuoka, et al. 1995), where $\mathbf{B}_{surf}$ and $\mathbf{B}_{tail}$ are functions of $r$, $\theta$, and $\phi$. Additionally, other models such as T96 and T01 (Tsyganenko 2001; Tsyganenko 1996) have incorporated symmetry in the magnetopause current and validated the differences in closure paths through satellite observational data.

And we have added additional description as:
**Line 280-283 (for equation 4) :** In our earlier study (*Gu et al.*, 2023), the magnetic field of the current system, denoted as $\mathbf{B}_{c0}$ (r), did not account for variations in $\theta$ and $\phi$. This limitation is addressed in the present study. The fundamental form of $\mathbf{B}_{c0}$ (r,$\theta$,$\phi$) , incorporating these angular dependencies, is introduced in Equation (4). The current system exhibits asymmetry effects, consistent with other models such as T96 and T01 (*Tsyganenko*, 1996; *Tsyganenko*, 2001). These models also incorporate dawn-dusk asymmetry in the magnetospheric current, reflecting the influence of these angular dependencies.

**Lines 363 – 382: Figure and the following discussion. Are the authors projecting the THEMIS position at the subsolar region? In this case would not be more interesting to calculate the magnetopause model results at the real THEMIS location? On my understanding, when the authors analyze the projection of the spacecraft position at the subsolar point, they are assuming the magnetopause is "shrinking" and could be neglecting any wave motion along the surface caused by a pressure pulse. Please, make a comment on that point.**

R: Thank you for raising this important point regarding the spacecraft positions. We clarify that for the statistical comparison of model predictions (Tables 3 and 4), the 38,887 THEMIS MCEs are not projected to the subsolar region (Line 315-316). The model results are calculated at the actual orbital locations of the spacecraft during each MCE, ensuring a direct and accurate comparison.

In the case studies presented in Figures 5 and 6, however, the spacecraft positions are projected. This projection is solely for visualization purposes, to improve the clarity of the figures when displaying multiple satellite observations and their relationship to the modeled magnetopause. This is a common practice in case study

presentations.

As you mentioned, the magnetopause's response to a pressure pulse can be complex, including in-phase, out-of-phase, and resonant motions, depending on the frequency of the pressure variations, as discussed in our previous work (Gu, et al. 2023). While an extensive discussion of this phenomenon would detract from the focus of this article, which is primarily model-oriented.

**Lines 425 – 426: Figure 6 and the following discussion. The authors claim that the POS model's predictions are more effective than C02 model. However, the results on Figure 6b are considering θ = 0 and the spacecraft locations are not. Also, in comparison with the satellite positions both models show very close results (by visual analysis) on figure 6C, the POS is closer on figure 6d and C02 is closer in figure 6e. I suggest the authors to point out these discrepancies.**

R: As you suggested, to provide a clearer comparison, we have labeled the errors in our case study and narrowed the plot range, thereby highlighting our model's capabilities more effectively (see Figure 6).

[Figure]

**Figure 1. A surface wave-like structure in X-Y magnetopause flank region. (a) The corresponding solar wind dynamic pressure(red) and IMF $B_z$ component(black), (b) The red, orange, yellow, green, blue, purple and black colours represent the initial magnetopause positions at φ =±80°, ±70°, ±60°, ±50°, ±40°, ±30°, ±20°, ±10°, 0°,**

respectively (dot line is the corresponding negative value of φ). The asterisk in purple (THEMIS-A), yellow (THEMIS-D) and red (THEMIS-E) indicate the satellite observation of MCEs projected onto the X-Y plane; (c) (d) (e) The shape of magnetopause in the X-Y plane at different time predicted by POS model (red dash line) and C02 model (green dot line), the asterisks represent the THEMIS MCEs positions mapped to X-Y plane.

**Discussion and Conclusion**

**How does the POS model respond to transient events formed at the bow shock or in the magnetosheath when they reach the magnetosphere? See Sibeck et al., (2022)https://doi.org/10.1029/2022JA030704 and Silveira C Sibeck (2023) https://doi.org/10.1029/2023JA031362 .**

R: Like many other magnetopause models, the POS model's response to transient events originating at the bow shock or in the magnetosheath is primarily driven by upstream solar wind observations (typically from L1 or pre-bow shock measurements). The POS model, in its current formulation, treats the magnetosheath as an intermediary region between the solar wind and the magnetosphere. Unlike many MHD simulations, POS model does not explicitly resolve the detailed physical processes and disturbances within the magnetosheath itself.

However, we have explored how the POS model responds to jets forming downstream of a quasi-parallel bow shock, as described by Silveira and Sibeck (2023a). In this scenario (illustrated in the figure below), the POS model predicts a magnetopause position (yellow) that seeming aligns well with MHD simulation (BATS-R-US). While the current model treats the magnetosheath as an intermediary, incorporating more detailed magnetosheath physics is a promising direction for future model development.

[Figure]

**Is POS model capable to catch return magnetic fluxes from nightside to dayside due to nightside magnetic reconnection? See Silveira et al., (2024)** https://doi.org/10.1029/2023JA032166**.**

**R:** The current version of the POS model focuses primarily on the interaction between the solar wind and the dayside magnetosphere and does not explicitly model the effects of nightside magnetic reconnection. Nevertheless, we have also investigated the case study in Silveira, et al. (2024a). As illustrated below, the current version of the POS model (yellow) predicts a magnetopause position that appears to align well with the MHD simulation (BATS-R-US), yet it shows a relatively gentle response to this phenomenon.

[Figure]

As with the previous comment regarding transient events in the magnetosheath, these two limitations highlight a potential area for future model development. Moreover, as the only time-dependent three-dimensional magnetopause model currently available, we are considering writing a separate paper in the future that compares the magnetopause's responses to various disturbances, as well as contrasting these responses with those predicted by MHD and other models.

Here, we have added the following proposals for improvement:
**Lines 476-478:** Moreover, magnetosheath transient effects and other perturbations on the magnetopause position are not addressed (Sibeck, et al. 2022; Silveira and Sibeck 2023b; Silveira, et al. 2024b).

**References**

Aubry, M. P., C. T. Russell, and M. G. J. J. o. G. R. Kivelson, Inward motion of the magnetopause before a substorm, (1970),75: 7018-7031

Boardsen, S. A., T. E. Eastman, T. Sotirelis, and J. L. Green, An empirical model of the high-latitude magnetopause, (2000),105(A10): 23193-23219

Chao, J., D. Wu, C.-H. Lin, et al. (2002a), Models for the size and shape of the Earth's magnetopause and bow shock, paper presented at Cospar Colloquia series, Elsevier.

Chao, J. K., D. Wu, C. H. Lin, et al., Models for the size and shape of the Earth's magnetopause and bow shock, COSPAR Colloquia Series (2002b),12: 127-135

Chen, C. X., and R. A. Wolf, Theory of thin-filament motion in Earth's magnetotail and its application to bursty bulk flows, Journal of Geophysical Research-Space Physics (1999),104(A7): 14613-14626

Collado-Vega, Y. M., P. Dredger, R. E. Lopez, et al., Magnetopause Standoff Position Changes and Geosynchronous Orbit Crossings: Models and Observations, Space Weather (2023),21(6): e2022SW003212

Dungey, J. W., Interplanetary Magnetic Field and the Auroral Zones, Physical Review Letters (1961),6(2): 47-48

Fairfield, D. H., Average and unusual locations of the Earth's magnetopause and bow shock, (1971),76(28): 6700-6716

Formisano, V., V. Domingo, and K.-P. Wenzel, The three-dimensional shape of the magnetopause, Planetary and Space Science (1979),27: 1137-1149

Gu, Y. X., Y. Wang, F. S. Wei, et al., Quasi-elastodynamic Processes Involved in the Interaction between Solar Wind and Magnetosphere, The Astrophysical Journal (2023),946(2): 102

Lin, R. L., X. X. Zhang, S. Q. Liu, et al., A three-dimensional asymmetric magnetopause model, (2010),115(A4):

Olson, W. P., The shape of the tilted magnetopause, (1969),74(24): 5642-5651

Plaschke, F., H. Hietala, and V. Angelopoulos, Anti-sunward high-speed jets in the subsolar magnetosheath, Annales Geophysicae (2013),31(10): 1877-1889

Shue, J.-H., P. Song, C. T. Russell, et al., Magnetopause location under extreme solar wind conditions, (1998),103(A8): 17691-17700

Shue, J. H., J. K. Chao, H. C. Fu, et al., A New Functional form to Study the Solar Wind Control of the Magnetopause Size and Shape, Journal of Geophysical Research (1997),102(A5): 9497

Silveira, M. V. D., and D. G. Sibeck, A Linear Velocity Gradient in the Subsolar Magnetosheath, Journal of Geophysical Research: Space Physics (2023),128(5): e2023JA031362

Silveira, M. V. D., D. G. Sibeck, F. R. Cardoso, and J. W. Gjerloev, Tracking the Subsolar Bow Shock and Magnetopause: Applying the Magnetosheath Velocity Gradient Method, Journal of Geophysical Research: Space Physics (2024),129(4): e2023JA032166

Wang, Y., and C. X. Chen, Numerical Simulation of Radial Plasma Transport in the Saturn's Magnetosphere, Chinese Journal of Geophysics (2008),51: 635-642